# Programming of Industrial Robots Using a Laser Tracker

**DOI:** 10.3390/s22176464

**Published:** 2022-08-27

**Authors:** Dariusz Szybicki, Paweł Obal, Krzysztof Kurc, Piotr Gierlak

**Affiliations:** Department of Applied Mechanics and Robotics, Faculty of Mechanical Engineering and Aeronautics, Rzeszów University of Technology, 35-959 Rzeszow, Poland

**Keywords:** industrial robots, robot programming methods, laser trackers, robot accuracy

## Abstract

This paper presents the use of a laser tracker for programming paths of industrial robots. The idea of operation, application, and the offered accuracy of the laser trackers are discussed. Then, the developed method of using a tracker for determining the path points of an industrial robot is presented. The requirements related to the developed method are indicated. The parameters of the industrial robot related to the repeatability and accuracy of the robots and the Absolute Accuracy option—increasing the accuracy—are characterized. The developed method is based on the accuracy of robots and the conversion of the position and orientation of individual coordinate systems. In the developed method, the robot is not used during programming and the programmer indicates the path points with the help of a retroreflector. The algorithm, which is the most important part of the developed method, has been implemented in the Rapid programming language to automate the programming process. This article provides an example of the use of the developed method to program the robot path during the grinding process. The advantages of the developed method are characterized and compared to existing solutions. In addition, the limitations and achieved accuracies are indicated.

## 1. Introduction

Newly installed robots, as well as those already in use, require the development of software or its modification, which is why robot programming methods are a rapidly developing field. The main purpose of the newly developed methods is to speed up the programming process and make it easier for staff to execute it. Reducing programming time is a source of cost savings and is of enormous importance to businesses.

Even 10 years ago, it was fairly easy to divide the methods of programming robots into on-line and off-line ones. Currently, there are methods that are difficult to clearly classify to one of these groups, so they are hybrid methods. A detailed description of individual methods of programming robots is described in [1,2]. The following programming methods can be distinguished:

### 1.1. Lead-Through Programming, On-Line Programming 

The lead-through programming [1,2] method consists of the fact that the operator carries out the movement of the robot’s arm using the manual panel, and the individual points of the robot’s trajectory are saved in the controller’s memory. It is the most popular method of programming industrial robots. In this method, the programmer, using his own senses, especially eyesight, has to drive the robot TCP (in accordance with the ISO 10218-1: 2011 tool center point—a point defined for a given application, taking into account the coordinate system of the mechanical interface) to the selected point using the joystick or the manual panel keys. This method is time consuming, relies on the precision of the programmer’s eyesight, and can cause collisions. Passing through the recorded points, the path of the robot’s tip is determined with the given accuracy and speed. This makes it possible to recreate the path set by the operator during the process execution. This method is used, for example, when CAD models of all essential elements of the robotic station do not exist.

### 1.2. Walk-Through Programming

One of the forms of programming robots is for the programmer to run the robot tip along the desired trajectory of movement and save the subsequent trajectory points in the memory of the robot controller [1,2]. Such a path is carried out by manually moving the tip of the robot equipped with a force sensor with a special grip. The force applied to the grip by the programmer is detected by the force measuring system and the robot moves the tip in the direction of the force. In the event of a moment force appearing, the robot rotates the tip in the direction of the moment. This method is used in the following cases: the desired robot motion paths are very complicated, and at the same time there are no environment models to program the robot off-line; it is required to program the robot to map the movements carried out so far by the human operator.

The benefit of this programming method is the maximum facilitation of this process and the possibility of its implementation by unskilled personnel, and easy adaptation of robots to the implementation of processes in the event of a change in the range of products.

An extension to walk-through programming is provided by the method of programming by demonstrating [1], whereby the robot is taught movements performed in various conditions and generalize them in new scenarios that have not been demonstrated before. A robotic station must have the ability to learn.

### 1.3. Off-Line Programming

The off-line programming method consists of generating robot paths with the use of a virtual environment [3,4]. CAD models of individual robots, auxiliary devices, such as feeders, tool changers, part warehouses, etc., are imported into the environment. It is also possible to use the CAD models of the designed elements. Then, robot paths are generated tracing the indicated contours of the CAD models or connecting the indicated points of the environment. The advantages of this method are the speed of generating paths, the safety of the programmer and hardware, and the convenience of the work. Apart from the advantages of this method, there are also limitations to programming. The basic and often overlooked problem in off-line design and programming is the absolute accuracy of industrial robots and the difference between accuracy and repeatability. Programming off-line requires high robot accuracy. The subject of examining the accuracy and repeatability of robots is the subject of many publications, and the methods of their determination are specified in, inter alia, ISO 9283: 1998. The accuracy of the robot is a measure of how close the robot can get to a given point (usually by coordinates) in the working space. Repeatability is a measure of how close a robot can get to a previously reached position. Robot manufacturers provide repeatability in their catalog cards and it is very difficult to obtain information about their accuracy. Experience shows that if the repeatability of the robot is, for example, 0.03 mm, the accuracy can be even several mm. One way to quickly and efficiently design and then program robots is to use online corrections. In this case, the coordinate systems, program points, and station logic are created off-line. Then, only those points or coordinate systems that require it are corrected on the real online workstation.

### 1.4. Programming with the Use of Augmented Reality

Augmented Reality technology belongs to the spectrum of virtual reality methods. Its basic feature is to integrate 3D computer-generated graphic objects into the real-world scene. This approach allows the programming of robots and robotic stations without the need to model all elements of the real environment [5]. It is enough, for example, to use virtual robot arms, whose CAD models are easily accessible, and immerse them in a real robotic station, in which there is a real object with which the robots interact [2]. In this way, robots can be programmed; e.g., for tracing the contours of a workpiece.

### 1.5. Programming of Robots Using Virtual Reality and Digital Twins

The essence of this method is the interaction of the operator with elements of the virtual environment. Its aim is to replace humans with robots in tasks whose formal description is complex [6,7,8]. It is usually used in cases where the sequence of movements needed to perform a given task has been experimentally selected by the operator using the method of multiple trials and on the basis of many years of experience. Such examples include grinding turbine blades, cleaning casting molds, painting processes, or complicated assembly of elements. This method is particularly advantageous when it is necessary to maneuver objects that in reality have, e.g., a large mass, as in a virtual environment, this is not a problem.

The most important methods of programming robots presented above have their advantages and disadvantages. As part of this research work, it was possible to develop a programming method showing some advantages over the methods presented here and using a laser tracker to indicate the robot’s TCP points.

This paper is targeted at people dealing with the subject of programming industrial robots working at universities, research institutes, or companies integrating robotic positions. The paper shows the methodology of using a laser tracker for programming with a real application example. The programming algorithm and the way of defining the necessary coordinate systems are presented. For people interested in robot programming, information on accessories that increase the accuracy of robots may be important, based on the example of the ABB Absolute Accuracy option (ABB Ltd., Zürich, Switzerland). The subject of this paper may also be interesting to people who already have laser trackers. They are usually used in metrology, accuracy, and repeatability studies of robots and other devices. This paper shows their interesting application in programming. Companies designing and programming industrial robots and considering purchasing a tracker for measuring robotic stations may, based on the conclusions of the paper, gain another argument related to its purchase.

A laser tracker is a device that allows measurements in three-dimensional space, used for probing, scanning, automated control, and reflector measurements. It is equipped with an absolute rangefinder and a laser interferometer mounted on a biaxial gimbal. Due to their precision, trackers are often used in studies of the accuracy and repeatability of robots [9,10,11]. An example of such a tracker is the Leica AT 960 head (Hexagon, Stockholm, Sweden) (Figure 1a).

This tracker allows for accurate measurement of the position of a selected point in three directions simultaneously. In the case of measurements on a robot, it is necessary to attach a mirror reflecting the laser beam. Measurement with a laser tracker is classified as coordinate-measuring systems. The principle of operation of trackers is based on the combination of two techniques, namely, laser interferometry, which allows measuring the distance of the target from the measuring head, and measuring the angles of setting two rotational axes: azimuth and height (Figure 1b).

In the case of laser trackers, mirrors are used as the measuring target to reflect the laser beam generated by the device. Most often these are Spherically Mounted Retroreflectors (SMR). The mirrors in the retroreflector are mounted precisely at the right angle to each other in such a way that their point of contact (apex) is exactly in the center of the SMR (Figure 2). This makes it possible to precisely determine the coordinates of the position of the retroreflector in three-dimensional space.

Laser trackers in robotic measurements are used for a variety of applications. Theissen, Laspas, and Archenti [12] present an innovative methodology for measuring the susceptibility of articulated serial robots, and a laser tracker is used to measure the response of the system.

Cvitanic, Nguyen, and Melkote [13] used a laser tracker to measure the deflection of the robot end effector during comparative tests and optimization of the robot position using static and dynamic stiffness models for various milling scenarios.

Nguyen and Melkote [14] investigated the modal properties of industrial robots, which change depending on the configuration of the arm in the milling process. A laser tracker was used as a measurement system for the position and orientation of the robot tool in three-dimensional space, to track the robot arm during milling.

Al Khawli et al. [15] presented a method of maintaining a high accuracy of robot manipulation through continuous compensation of the position errors in production processes in the aviation industry. Continuous tracking of the position and orientation of the mounted tool on the robot arm minimize the errors between the tool and the workpiece. For this purpose, among others, the Leica Absolute Tracker laser tracker and the Leica Tracker-Machine Control (T-Mac) (Hexagon, Stockholm, Sweden) reflector were used to record measurements relative to the base of the robot.

Novák et al. and Mei et al. [16,17] considered how to increase the accuracy of industrial robots with the help of the Leica Absolute Tracker AT960 (Hexagon, Stockholm, Sweden). They propose new methods of calibrating robots with tools in their workplace. These methods improve the positioning accuracy by compensating for the identified parameters. The accuracy of the robots, along with the reduction in calibration time, are key factors in the success of robotic production systems.

Fernandez, Olabi, and Gibaru [18] discussed assembly operations in the aviation industry, which are time-consuming and require high accuracy. They emphasized that robotic assembly is a good solution that increases productivity, but pointed out that the poor accuracy of industrial robots limits their use. They proposed an improvement by adding an accurate on-line 3D positioning system, which consists of the KEYENCE LJ-V7200 vision system (Keyence, Osaka, Japan) and the Leica AT-960 + T-Mac TMC-30B (Hexagon, Stockholm, Sweden) tracking system.

Slater et al. [19] presented a new Portable Laser Guided Robotic Metrology (PLGRM) system at the National Aeronautics and Space Agency (NASA) for robot positioning and displacement. This system consists of a cooperating robot arm mounted on a lift and a laser tracker located on a movable base. Together, they allow for scanning an area larger than the range of the robot.

Gonzalez et al. [20] focused on measuring the quasi-static path accuracy and repeatability of industrial manipulators to evaluate their performance in industrial contact applications such as trimming, grinding, or deburring. The validation was performed on an ABB industrial robot using a Leica AT960 laser tracker.

Sanden, Pawlus, and Hovland [21] investigated the ability of a laser tracker to measure the relative position and orientation between two mobile Stewart platforms simulating the movement of ships at sea. These ships are exposed to disturbance from waves and have cranes equipped with active compensation systems on board, which keep the cargo at a certain height from the seabed.

Martin et al. [22] used a laser tracker to improve the accuracy of cable-driven parallel robots. Inaccuracies are caused by deviations in cable lengths caused by elongation, elasticity, or creep.

Theissen, Mohammed, and Archenti [23] focused on modeling, measuring, and identifying the change in the kinematic chain of serial articulated industrial robots based on thermomechanical deformations caused by the self-heating caused by drives. The assessment of the change in the positioning accuracy of the ABB IRB 1600 (ABB Ltd., Zürich, Switzerland) robot was carried out using a Leica AT960 laser tracker and a FLIR SC640 (FLIR, Wilsonville, OR, USA) thermal imaging camera.

Laser distance measuring systems are widely used, not only in industry. Yu, Li, Guan, and Wang [24] used the RIEGL VMX-450 (Riegl, Horn, Austria) system to test a new algorithm to quickly extract objects from urban road spaces, such as lighting poles, road signs, and bus stops. The system uses an array of laser scanners and cameras to map the space around the device. It is equipped with a satellite navigation system and a device for measuring the displacement of the wheels of the vehicle on which it is mounted.

Laser scanners are also often used in autonomous vehicles. Li et al. [25] proposed the use of the LIDAR 3D system in the VFH (Vector Field Histogram) algorithm of a local path planner (generator) for the navigation system of autonomous vehicles.

Another approach to tracking objects in space are vision systems, which use special tags that allow the positions and orientations of an object to be determined. An interesting example of such an approach is presented by Ferreira, Costa, Rocha, and Moreira [26], who built a marker in the shape of an icosahedron. LEDs in five different colors are placed on the walls in such a way as to obtain a unique pattern for each of the walls. The moving marker is tracked by a stereoscopic vision system. This system was used to teach robots painting movements.

As indicated, laser trackers have a wide range of applications in robotics; however, after researching the market and browsing articles and websites, no applications of the tracker for generating (programming) paths for industrial robots were found.

## 2. Description of the Robotic Station

The paths of an industrial robot were generated with the use of a laser tracker at a robotic station located in the laboratory of the Department of Applied Mechanics, Rzeszow University of Technology (Figure 3).

The robotic station includes:An ABB IRB 2400 (ABB Ltd., Zürich, Switzerland) (number 1) industrial robot with the Absolute Accuracy option, addition of force control, and a tool changer; optionally, a 2.2 kW electrospindle, a two-finger parallel gripper, a 2D scanner, a 3D scanner of structured light, and various types of pneumatic tools can be installed.An ABB IRBP A250 (ABB Ltd., Zürich, Switzerland) (number 2) two-axis positioner cooperating with robots and capable of assembling workpieces weighing up to 250 kg.An Absolute Leica AT960 (Hexagon, Stockholm, Sweden) (number 3) laser tracker complete with accessories.

Among the elements of the station, the parameters related to the accuracy of the robot and the Absolute Accuracy option as well as the accuracy of the tracker are important in relation to the topic of this article.

### 2.1. Description of the Leica Tracker

The accuracy specification of the given laser tracker was given in the form of the maximum permissible error (MPE). The maximum permissible error (MPE) is defined by the ISO 10360-10: 2016 standard as the highest measurement error value allowed in the specification for a given measurement; i.e., the length measurement. An acceptance test is confirmed if all measurement errors according to the test guidelines are less than the corresponding MPE specifications. The ISO 10360-10: 2016 standard defines the criteria for repeating measurements in excess of the MPE values. Typical Absolute Tracker measurement results are half of the corresponding MPE values. Table 1 shows the specifications of the laser tracker subsystem according to ISO 10360-10: 2016, Annex E [27].

Table 2 shows the MPE specifications for the tests defined in ISO 10360-10: 2016 when using 1.5″ Leica Red Ring Reflectors (RRR) and the Standard Test Mode, unless otherwise stated.

The transverse angle, eT, was according to ISO 10360-10: 2016, related to the MPE for the localization error (LDia. 2 × 1: P&R: LT, MPE), according to Chapter 6.3 of ISO 10360-10: 2016, 30 μm + 12 μm/m.

### 2.2. Description of the Robot and the Absolute Accuracy Option

ABB IRB 2400 is a general-purpose robot with a payload of 16 kg, a position repeatability of 0.3 mm, and a path repeatability of 0.15 mm. It has the Absolute Accuracy option, which is important in the context of the topic of the paper. This option is used to increase the absolute accuracy of the robot and includes compensation for mechanical properties and deflection under load. Depending on the model, the difference in accuracy between the robot model and the real robot is 8–15 mm. The use of the Absolute Accuracy option guarantees an accuracy of approximately 0.5 mm across the entire workspace. The compensation of robot accuracy errors due to mechanical tolerances and deflections under load is a complex and highly non-linear problem. In the case of ABB, there is a set of around 40 parameters that describe the individual properties of a calibrated robot. The positioning accuracy of a calibrated robot depends on the size and variant of the robot, and averages from 0.25 to 0.55 mm for a robot handling 5 to 500 kg. Other robot manufacturers also have add-ons or options similar to Absolute Accuracy. Their functioning and the obtained parameters are similar.

The addition of this option is made for each manipulator individually, and it is usually done at the factory. The CalibWare calibration software (ABB Ltd., Zürich, Switzerland) and procedure requires a robot to be measured and a dedicated test station is needed for this. ABB uses a Leica laser tracker to perform the calibration measurements. Calibration consists of driving to approximately 100 random points in the work area. As a result of the calibration, a set of parameters is calculated that minimizes the error between the model and the real robot. Ultimately, the robot’s positioning accuracy is verified in 50 additional positions by calculating the differences between the given position and the measured position, as shown in Figure 4.

The accuracy of each robot will be confirmed and verified through a “birth certificate” that statistically describes the accuracy of the robot in a large sample of positions. The most important information from the certificate of accuracy issued for our robot is given in Table 3. The measurement was performed with a Leica AT901B tracker with a maximum error of 10 µm/m.

Owning a robot with the Absolute Accuracy option with a maximum accuracy error of 0.38 mm and a laser tracker, a method of programming the robot was developed using measurements made with the tracker.

## 3. The Developed Programming Method

For the implementation of the industrial robot programming concept, the procedure algorithm shown in Figure 5 was developed. After the implementation of this algorithm, the programming process consists only of indicating with the retroreflector the points that the robot’s TCP is to reach. During the development of the algorithm, it was assumed that after selecting a point with a retroreflector, the process of determining the path points is to be automated. All the calculations necessary to implement the algorithm are to be performed on the robot controller in the RAPID language (ABB Ltd., Zürich, Switzerland).

The first stage of the robot programming process with the use of a tracker is to determine the position and orientation of the base system, the so-called Base Frame of the robot. In the case of ABB robots, this system is the basic and most important one. In relation to it, the position and orientation of the TCP are determined, the positions and orientations of path points are determined, and other coordinate systems are defined. The location of this system for our ABB IRB 2400 robot is shown in Figure 6.

As shown in Figure 6, the base system in the base of the robot is in a restricted area and it is not possible to physically indicate this point. Its position is fixed in relation to the mounting bases, so its determination would require disassembling the robot. It was decided to use a different method of determining the mutual position and orientation of the systems: the base manipulator and the tracker. If the robot has the Absolute Accuracy option, it is possible with a satisfactory precision of 0.38 mm in the worst case to determine the robot’s TCP system in relation to the base system. Determination of a common frame of reference for both devices is performed by mounting the retroreflector in the robot TCP (Figure 7). A minimum of three retroreflector positions relative to both systems (robot base and Leica systems) are required. The calculation of the transformations was performed on the robot controller using the functions of the RAPID language. Details of the transformation in RAPID language can be found in the file attached in Appendix A.

The position of the retroreflector mounted in the robot’s TCP was determined in relation to the Leica W_L_ system and the same position was read from the robot control system in relation to the W_B_ base system. This operation was repeated for three TCP positions. The decision was made on the 3rd position, because three points can be used to build a coordinate system with a specific position and orientation. The positions are written as vectors containing the coordinates of the points in the adopted frame of reference.
(1)$$T_{1}=\left[\begin{array}{c} x_{T1}\\ y_{T1}\\ z_{T1} \end{array}\right],T_{2}=\left[\begin{array}{c} x_{T2}\\ y_{T2}\\ z_{T2} \end{array}\right],T_{3}=\left[\begin{array}{c} x_{T3}\\ y_{T3}\\ z_{T3} \end{array}\right]$$

This data is stored in the pose variable (Rapid language data type). Then, using the Rapid—*DefFrame* language instruction, the W_T_ coordinate system defined in relation to the base system was created from the obtained three points, T_1_, T_2_, and T_3_. This instruction creates a coordinate system with origin at point T_1_, and the x_T_ axis passes through point T_2_. The x_T_y_T_ plane includes all three points, T_1_, T_2_, and T_3_, and the z_T_ axis is obtained from the definition of a right-handed system, the concept of this function is shown in Figure 8.

The function calculates the position of points T_3_ and T_2_ in relation to point T_1_ as
(2)$$r_{T2-T1}=T_{2}-T_{1}$$
(3)$$r_{T3-T1}=T_{3}-T_{1}$$

Then, the unit vector i_T_, describing the position of the x_T_ axis in relation to the base system, is calculated:(4)$$i_{T}=\frac{r_{T2-T1}}{||r_{T2-T1}||}$$

The unit vector k_T_ is formed as the following vector product:(5)$$k_{T}=\frac{r_{T2-T1}\times r_{T3-T1}}{||r_{T2-T1}\times r_{T3-T1}||}$$

Knowing the position of the x_T_ and z_T_ axes, the unit vector j_T_ can be calculated as the vector product in the form
(6)$$j_{T}=\frac{k_{T}\times i_{T}}{||k_{T}\times i_{T}||}$$

The matrix of rotation of the W_T_ to W_B_ system (Figure 9) will take the form
(7)$$R_{B}^{T}=\left[\begin{array}{ccc} i_{T} & j_{T} & k_{T} \end{array}\right]$$

The starting point of the W_T_ system is point T_1_, so the system’s shift vector will be
(8)$$d_{B}^{T}=T_{1}$$

This will allow us to write the homogenous transformation of the W_T_ to W_B_ system,
(9)$$H_{B}^{T}=\left[\begin{array}{ll} R_{B}^{T} & d_{B}^{T}\\ 0 & 1 \end{array}\right]$$
which explicitly determines the position and orientation of the W_T_ system in relation to the robot’s base system. The position and orientation of the W_T_ system in relation to the W_L_ were determined in the same way:(10)$$H_{L}^{T}=\left[\begin{array}{ll} R_{L}^{T} & d_{L}^{T}\\ 0 & 1 \end{array}\right]$$

Thanks to these transformations, it is possible to convert position measurements from the tracker to the robot’s base system. The following conversion is used for this:(11)$$P_{L}^{B}=H_{B}^{T}H_{T}^{L}P_{L}$$
where $H_{T}^{L}$ is the transformation of the tracker system to the W_T_ system, which is obtained by inverting the transformation described by the Equation (10),
(12)$$H_{T}^{L}={\left(H_{L}^{T}\right)}^{-1}$$
and P_L_ is the vector of the measured values with respect to the tracker system.

The Rapid language includes ready-made instructions for computing datum transformations. The *PoseInv* instruction was used to reverse the frame of reference. Then, the ready *PoseMult* instruction was used, which is used to calculate the product of two transformations. It determines the position of the system, as shown in the diagram in Figure 9.

The idea behind this instruction is as follows:


*VAR pose pose1;*



*VAR pose pose2;*



*VAR pose pose3;*



*…*



*pose3:= PoseMult(pose1, pose2).*


Thanks to this operation, it was possible to obtain the position and orientation of the Leica W_L_ system in relation to the W_B_ base frame. Using the *PoseInv* instruction, an inverse was also found; that is, the position and orientation of W_B_ with respect to W_L_. For greater precision in determining the mutual position of the systems, this operation was repeated for three three-element TCP positions in different areas of the workspace. This was to determine the position of the underlying system with greater precision.

Knowing the mutual position of the W_L_ and W_B_ systems, it is possible to measure the position of the point in the Leica W_L_ system and the transformation of this position to the W_B_ base system. Due to the fact that the laser tracker is mobile and is often used for other measurements, its position and orientation may change. Changing the position and orientation of the tracker would require a new procedure for determining the position of the base system and the implementation of the algorithm from Figure 5. When performing calculations on the robot’s controller, the transformation values are saved directly in its memory, which allows for automating the calibration process.

To speed up and facilitate the process of programming the robot with the help of a laser tracker, it was decided to define a fixed coordinate system that is stationary with respect to the base system. This system, unlike the base system, is available for the tracker to measure. Thanks to the use of dedicated retroreflector holders, the position of this system can be defined quickly and in a very repeatable manner (Figure 10). The dedicated reflector holders used ensure the repeatability of the retroreflector holder of 0.005 mm.

After attaching the retroreflector holders, as shown in Figure 10, the coordinate system in the robot pedestal W_P_ was defined in relation to the Leica work object W_L_ (as described previously from the three points). Then the position (x = 13.69, y = −311.42, z = 27.50) and the orientation of the coordinate system in the robot pedestal W_P_ in relation to the W_B_ base system were determined (Figure 11).

Thanks to this approach, when programming the robot, it will not be necessary to determine the position of the W_B_ base system in the manner described earlier (Figure 5); i.e., by mounting the retroreflector in the robot’s TCP. It is usually difficult because a dedicated tool, e.g., an electrospindle, is installed in the TCP. The controller software has been designed in such a way that before starting the point programming, the retroreflector should be applied to the holders and the system will automatically determine the position and orientation of the coordinate system in the robot pedestal W_P_. The position of the W_P_ system in relation to W_B_ is assumed as a constant in the software of the robot controller. Having determined the mutual positions of the W_L_, W_P_ and W_P_ systems in relation to the W_B_, it is possible to proceed to the robot programming procedure.

## 4. Functioning and Tests of the Developed Programming Method

The procedure of programming the robot after the implementation of the algorithm in Figure 5 is quite simple. It consists of bringing the retroreflector to the places that the robot’s TCP is to reach. As an example, the process of grinding the edge of a steel rim (Figure 12a) with the use of an electrospindle with an abrasive tool was selected. To speed up the programming process, the retroreflector was mounted on a dedicated stand defined in the Leica measurement system (Figure 12b). This made it easier to move the retroreflector around the edges and the measuring points were automatically recalculated by the tracker software.

Another facilitation during programming was the selection in the tracker software of the option of automatic recording of the measuring point at every distance; in this case, it was 50 mm. Thanks to the described software capabilities, generating the machining points (Figure 13) was limited to moving at a low speed along selected edges and it took about 40 s.

Figure 14 shows the station model, detail, and generated path points.

During programming, it was assumed that the robot tool, i.e., TCP bound with the abrasive wheel, was defined correctly.

The robot programming process is not only limited to defining the TCP coordinates at subsequent points in the path. It should be indicated, inter alia, the tool orientation during the process, type of motion interpolation, robot arm configuration, and speed. The selected process type (edge grinding) causes the electrospindle to be oriented perpendicularly to the xy plane of the base system during the process. This type of setting defines the orientation of the tool. When it comes to interpolation of motion, in this process, as in the vast majority of others, we are dealing with the linear motion instructions *MoveL* and circular motion instructions *MoveC*. These instructions operate on the same *Robtarget* variables. In the case of the implemented process, the movement in a circle was carried out using the *MoveC* instruction and commuting with *MoveL*. An automatic determination procedure was used to determine the robot’s configuration for individual positions. During functional testing, the robot reaches the first point of the path, displays a message whether the current configuration is correct, and, if so, it selects it or the closest for the subsequent path points. The TCP speed for the selected process results from the tool type and the type of machining and has been defined at 500 mm/s for all points.

Figure 15 shows a communication diagram of the Laser Tracker—ABB Robot. The controllers of both devices can communicate using the TCP/IP protocol family and the Ethernet standard. At the present stage, it is not possible to connect both controllers directly due to programming and hardware limitations. A PC with an application that mediates the exchange of information between devices is required.

The order of data transfer is as follows:The tracker software installed on the PC saves the positions of three points on the robot pedestal to a text file, then any number of points on the robot’s path;The robot controller automatically downloads and reads a text file containing the coordinates of the points on the robot pedestal (the first three) and the path points;The robot controller software, written in Rapid, determines the coordinates of the necessary coordinate systems;The coordinates of track points from the tracker are converted to coordinates relative to the base system;The orientations and configurations of path points are determined;The operator assigns the defined position variables to the selected instructions of the linear type *MoveL* and circle *MoveC*;The program with laser tracker points is ready for testing.

After the development of this software, the role of the programmer is limited to determining the path points with the help of a retroreflector. Marking points is facilitated by dedicated tracker stands for determining edges, holes, planes, etc. The tracker software options, such as automatic generation of a point at a specified distance or time, are also useful.

## 5. Advantages, Disadvantages, and Errors of the Developed Method—Discussion

As with any programming method, this one also has advantages and disadvantages and is characterized by the occurrence of typical errors.

The main advantages of the developed method include:High speed and time saving of the programmer;When using dedicated holders for a retroreflector, it is easy to determine edge points, surfaces, hole centers, etc.;Safe for the programmer and the robot because the robot is not used during programming, minimizing the risk of collisions resulting from the error of manual manipulator movements;Using the robot controller software or tracker software, it is possible to define new points (so-called virtual, difficult to indicate or reach with a robot on-line) on the basis of the already existing ones, e.g., the center of a circle based on three points, points on the extension of an already defined edge, etc.The disadvantages of the developed method are:The need to have an expensive tracker and an Absolute Accuracy option in the robot;The position of the path points is determined with an accuracy of 0.38 mm, which is a worse value than in the case of on-line programming (then we use repeatability, i.e., 0.03 mm);The robot TCP orientation at defined points must be determined by another method (e.g., in the described case perpendicular to the xy plane of the base system), although this problem has been solved by applying the measuring probe to the tracker, which will be the subject of further research;Before using this method for the first time, it is necessary to determine the robot’s base system in accordance with the algorithm (Figure 5).

To properly evaluate the proposed method of programming industrial robots, it should be compared with the two most-popular programming methods, namely, on-line programming and off-line programming. The comparison of methods and individual criteria are included in Table 4.

The criteria in Table 4 allow for the comparison of the proposed programming method with other commonly used methods. It should be noted that precise quantification of methods is only possible on a case-by-case basis. Programming time and costs depend, among others, on the complexity of the position and the number of path points. The requirements for programming a straight path in the palletizing process will be different, and the programming of the advanced process of deburring an aircraft part will look different. A good example for comparison is the process of grinding the edge of a steel rim shown in Section 4. It took about 40 s to create about 35 path points on the workpiece with the proposed method. To this time, the time of setting the tracker and determining the base system should be added. Having a defined layout of the pedestal, the time of setting the tracker and determining the necessary systems is about 300s. The total time to create 35 points on a part is about 340 s. In the case of the on-line programming method, having a properly defined tool, the programming time for the same points for a skilled programmer is about 15 min; i.e., 900 s. In the case of the off-line programming method, a CAD model of the rim and the position is necessary to create the same 35 points of the path. In the discussed case, the stand model was accurate, but the rim was a welded element and its CAD model was only close to reality; also, the machined rim edges were uneven. Working on the given CAD model, the creation of points is quick and took about 60 s. After the points were made, they would have to be transferred to a real robot controller and calibrated; this would take about 250 s. Comparing the programming time, the proposed method is comparable to the off-line programming method and three times faster than on-line programming. However, it should be remembered that for off-line programming, it is necessary to have an accurate CAD model of the workpiece, and when using a tracker, any points on the real workpiece are programmed.

The errors of the developed method result primarily from the accuracy of determining the coordinates of points with the use of a tracker and the accuracy of the robot determined by the Absolute Accuracy option. The accuracy of the tracker is very high compared to that of the robot. Therefore, the error of the method is mainly due to the accuracy of the robot and is around 0.4 mm. To confirm the assumptions about the error, a retroreflector was installed in the robot TCP (tool holder Figure 16). The position of the thus assumed TCP was determined in the robot system by means of a tracker with an accuracy of 0.05 mm.

The next stage was to test the robot’s movement to the defined points and the comparison of the position determined by the robot’s system and the tracker. Figure 17 shows the error plots on the individual axes. The error was determined as the absolute value of the difference in coordinates determined by the laser tracker and the robot system. Measurements were made for 10 points in different areas of the workspace.

The maximum error was noted for Measurement 4 on the z axis, amounting to 0.25 mm. The obtained value is less than the expected maximum error, possibly because the selected points are in the useful workspace of the workpiece. It is impossible to check all the possible robot positions. It was decided to measure the error in determining the position in a way more similar to the conditions in which the developed method will be used. For this purpose, a test was carried out on a workpiece on a robotic stand (Figure 18). During the test, points in the working space on the workpiece were determined.

A dial gauge with digital readout was installed and defined in the robot TCP. The dial gauge used was a precise device that measures in one axis with an accuracy of 5 µm. The TCP of the sensor was determined at the indication of 1 mm, with the accuracy resulting from the repeatability of the robot (0.03 mm). For the station prepared in this way, reference points were determined by means of a retroreflector and the developed system (Figure 19), which is to be achieved by the robot.

To determine the points, dedicated Leica tracker stands were used, intended to be indicated by the edge retroreflector. After determining the points, the algorithm developed in this article was used (Figure 5) and the TCP of the robot reached the defined points (Figure 20).

To obtain more reliable measurements, the points defined by the retroreflector were placed on mutually perpendicular planes. The obtained measurement results are 1.296 mm and 1.118 mm, respectively. Taking into account the initial sensor reading (1 mm), the obtained error can be defined as 0.296 mm and 0.118 mm. The error obtained is less than that determined by holding the retroreflector in the robot TCP. When analyzing these errors, it should be remembered that the dial gauge gives the measurement in only one axis. Detailed research on errors in the developed method will be the focus of subsequent work.

## 6. Conclusions

The paper describes the most-popular methods of programming industrial robots, indicates the main differences between them, and presents their advantages and disadvantages. A new programming method based on the use of a laser tracker was proposed. Describing the developed method, it was necessary to explain the idea of the operation of laser trackers, their applications, and the options ensuring the absolute accuracy of the robots. The idea of the developed method was described in the following sections. It was indicated how to prepare the robot system for cooperation with the tracker, and how to define and translate the necessary coordinate systems. The functioning of the prepared software in the Rapid language and the way of communication between the tracker controller and the robot controller were described. The last but very important part in this paper was to indicate the advantages and disadvantages of the developed method and to provide a brief description of the errors that occur. The method of measuring the error of about 0.4 mm was described and examples of errors occurring in the conditions of using this programming method were shown. Comparing the developed programming method to the currently most-popular on-line programming method, using a manual panel, it is possible to indicate its great advantage in terms of programming speed and programmer safety. The developed method was developed taking into account several aspects: the use of a tracker to determine the TCP of a robot tool; the construction of digital twins with the use of a tracker; and the definition of path points with the use of a dedicated measuring probe.

## Figures and Tables

**Figure 1 sensors-22-06464-f001:**
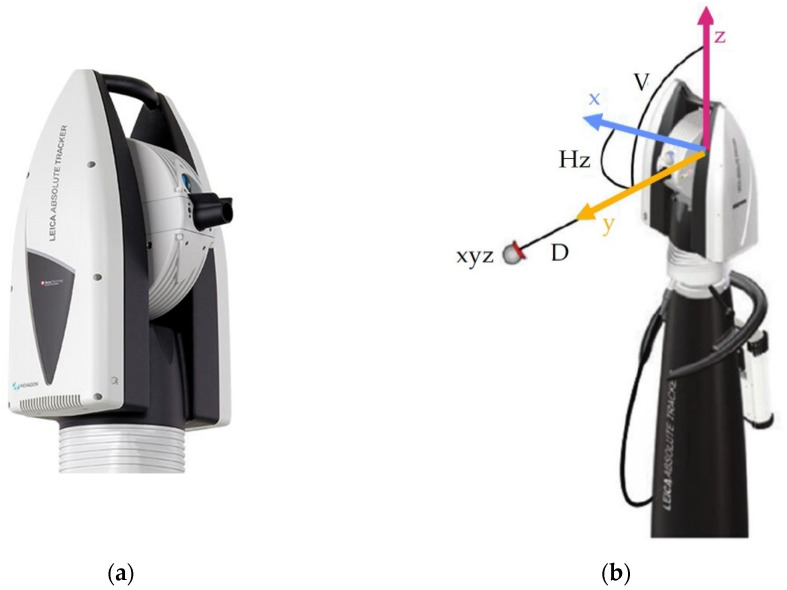
Laser tracker: (**a**) Leica AT 960 head; (**b**) the concept of measuring with a laser tracker.

**Figure 2 sensors-22-06464-f002:**
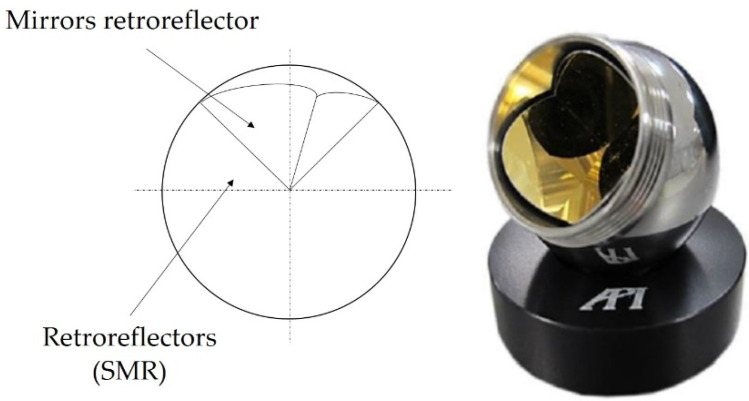
Construction of a laser tracker retroreflector.

**Figure 3 sensors-22-06464-f003:**
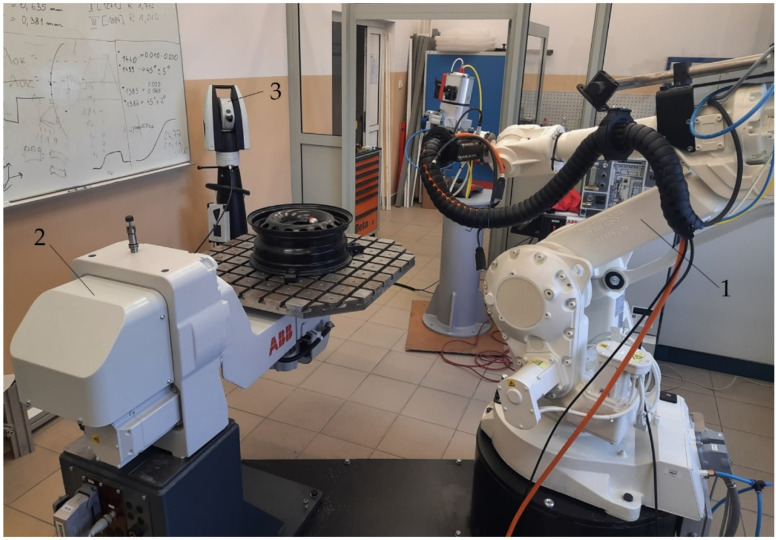
The robotic station: (1) ABB IRB 2400 robot, (2) ABB IRBP A250 positioner, (3) AT960 laser tracker.

**Figure 4 sensors-22-06464-f004:**
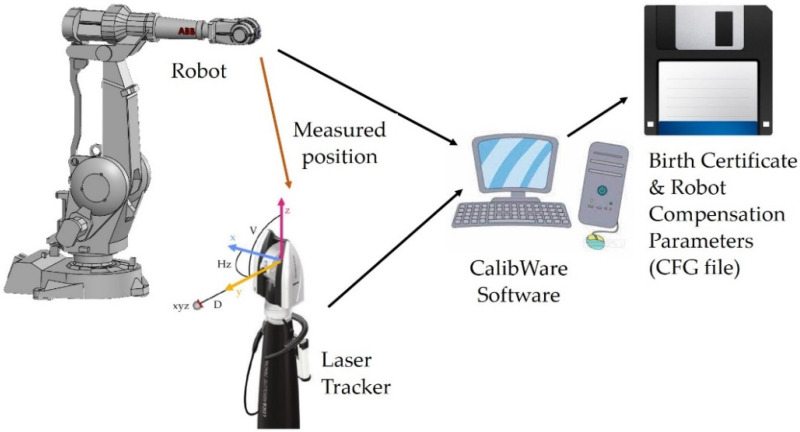
Configuration of the ABB calibration process.

**Figure 5 sensors-22-06464-f005:**
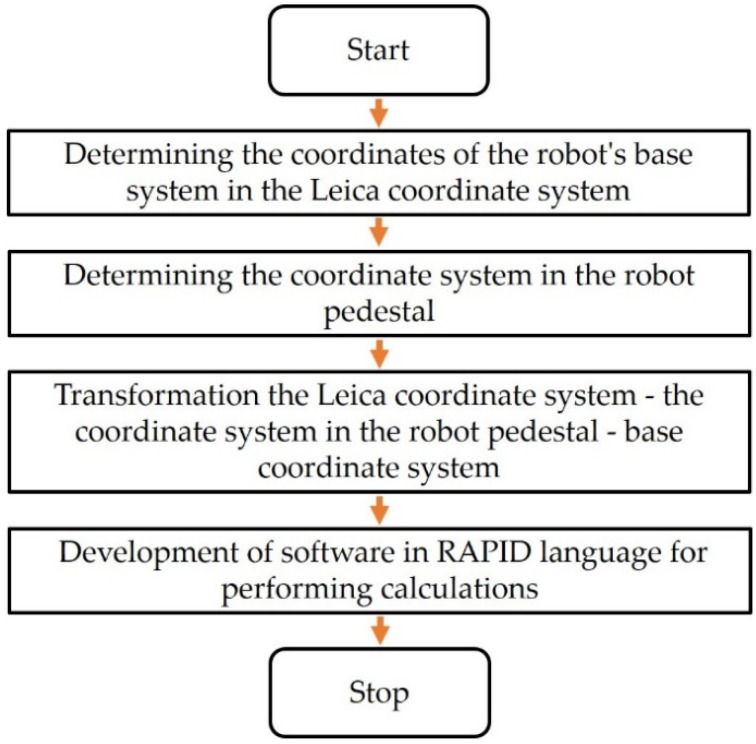
The proposed algorithm of the robot programming process.

**Figure 6 sensors-22-06464-f006:**
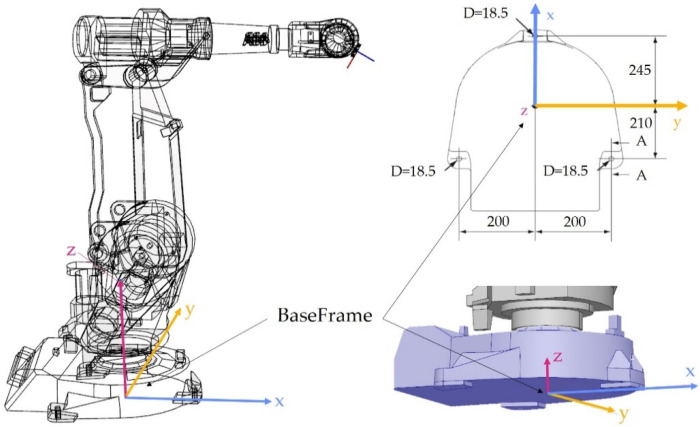
Position of the Base Frame in the ABB IRB 2400 robot.

**Figure 7 sensors-22-06464-f007:**
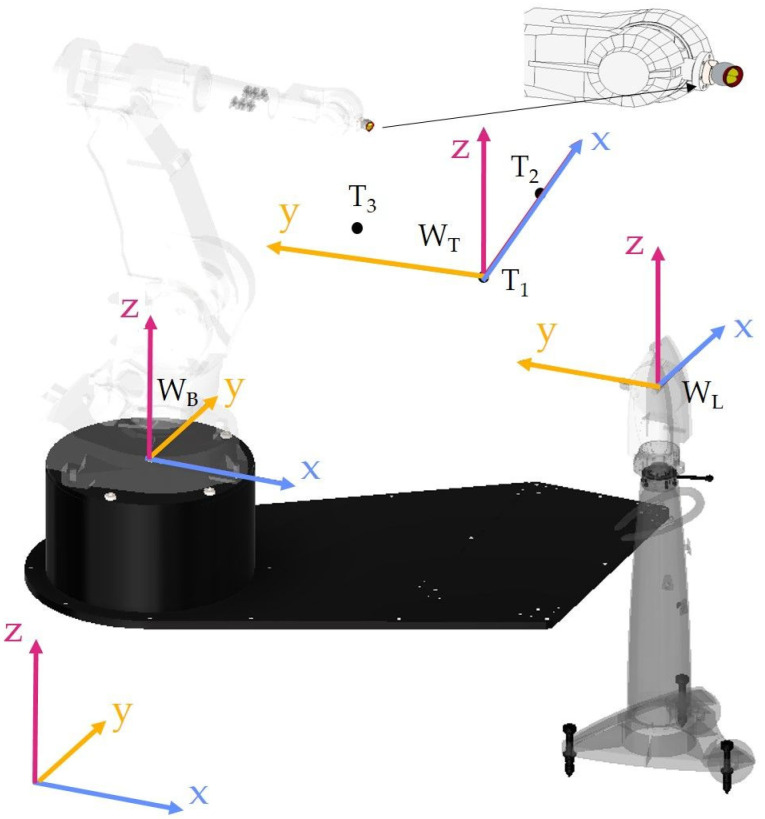
The idea of determining the Base Frame.

**Figure 8 sensors-22-06464-f008:**
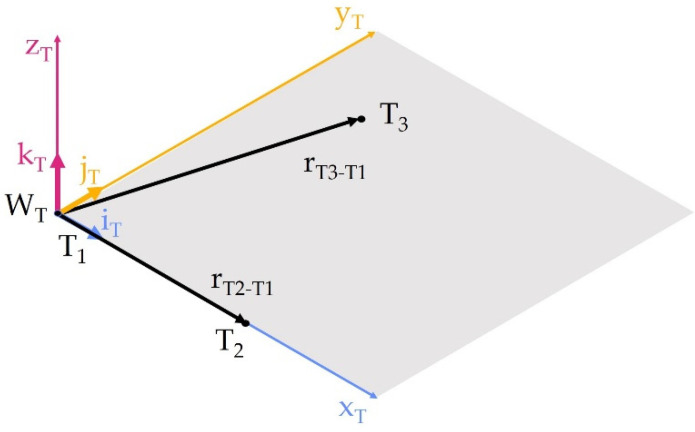
Determination of the W_T_ coordinate system.

**Figure 9 sensors-22-06464-f009:**
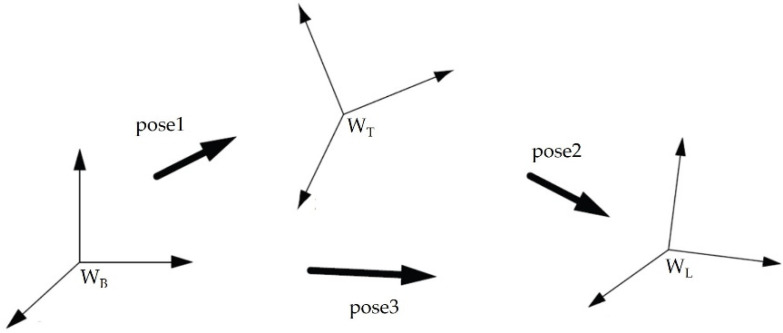
The idea behind the *PoseMult* instruction.

**Figure 10 sensors-22-06464-f010:**
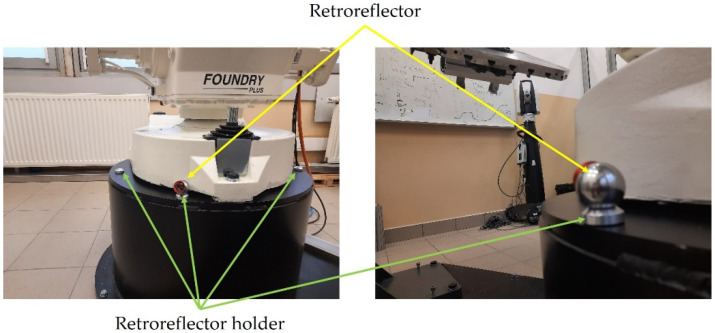
The location of the retroreflector holder.

**Figure 11 sensors-22-06464-f011:**
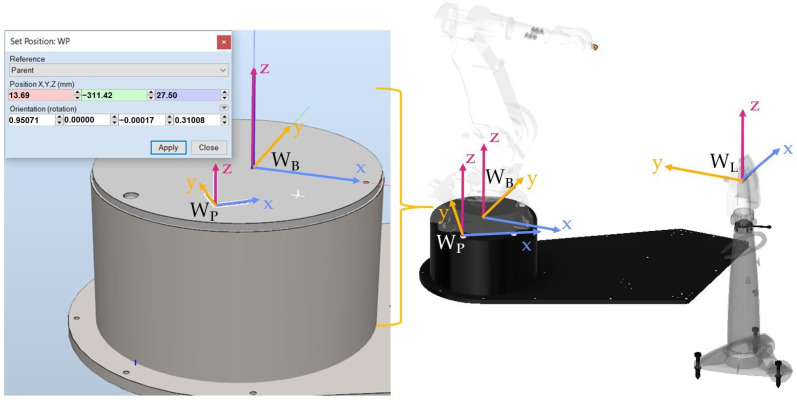
Position and orientation of the coordinate system in the robot pedestal W_P_ in relation to the W_B_ base frame and the position of the Leica W_L_ system.

**Figure 12 sensors-22-06464-f012:**
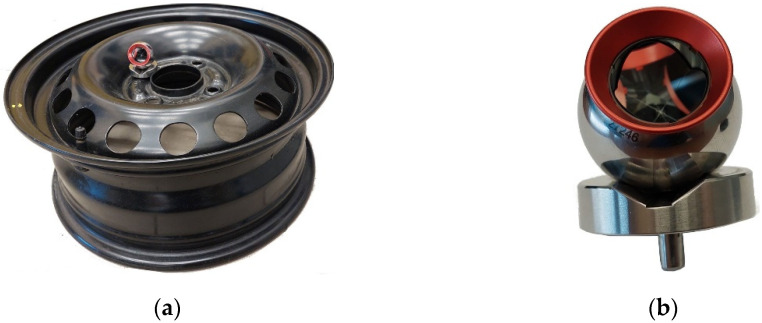
Elements used during tests: (**a**) detail; (**b**) retroreflector in a dedicated stand.

**Figure 13 sensors-22-06464-f013:**
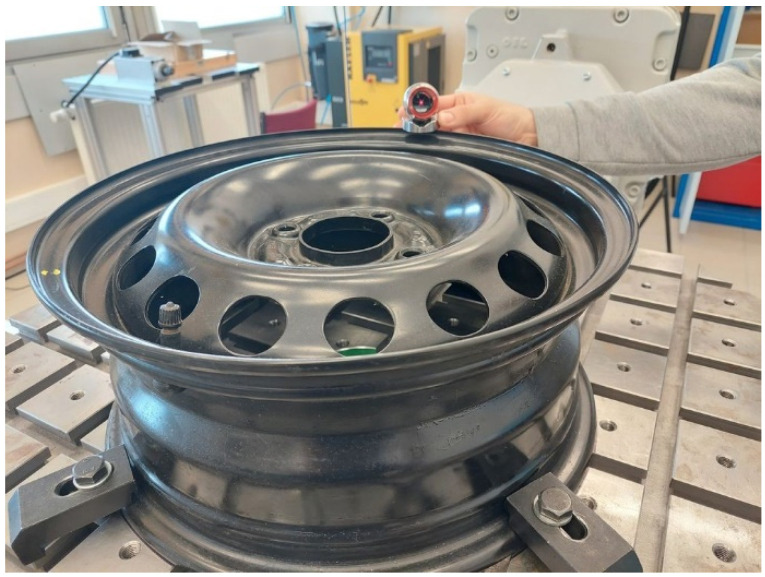
Retroreflector in the holder and detail during programming.

**Figure 14 sensors-22-06464-f014:**
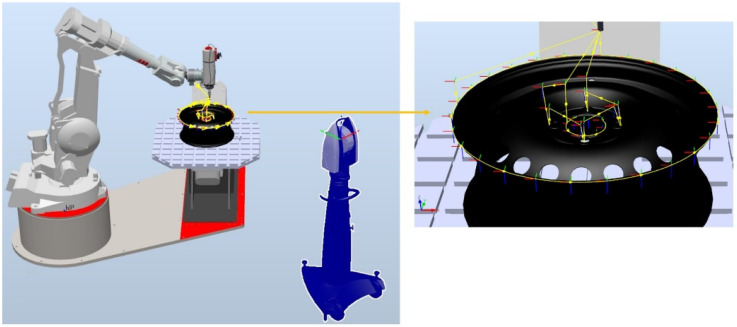
Station model in RobotStudio along with the detail and generated points.

**Figure 15 sensors-22-06464-f015:**
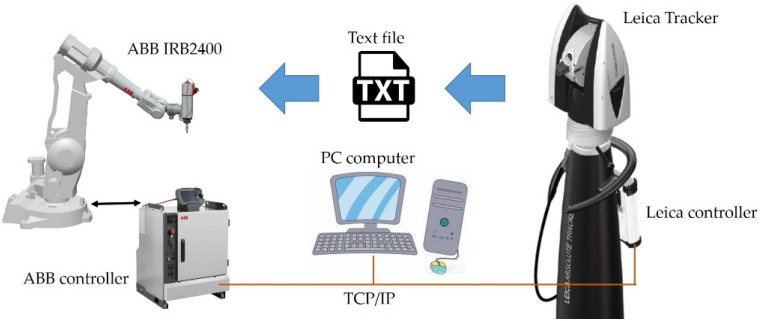
Communication diagram of the Laser Tracker—ABB Robot.

**Figure 16 sensors-22-06464-f016:**
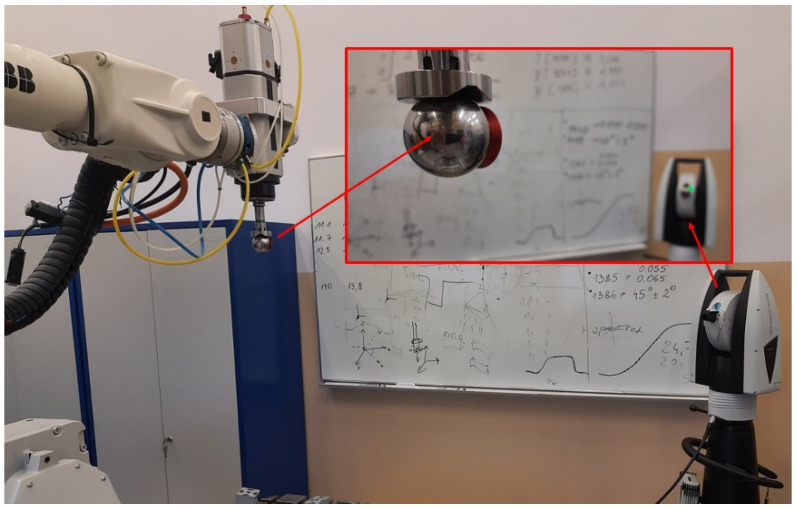
Retroreflector in the tool holder and the laser tracker.

**Figure 17 sensors-22-06464-f017:**
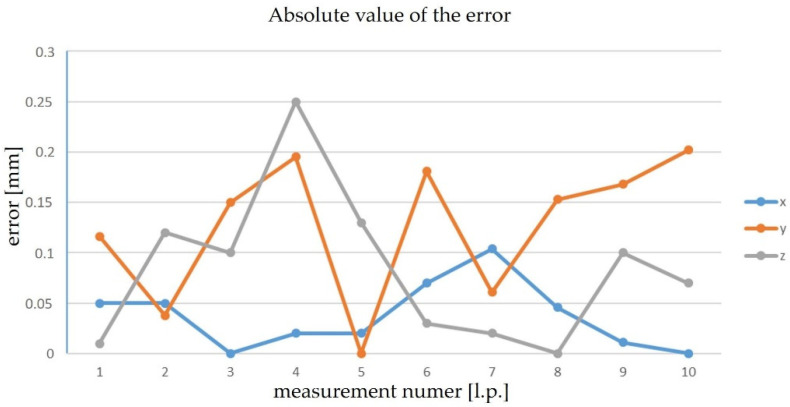
Error plots of the coordinates of points.

**Figure 18 sensors-22-06464-f018:**
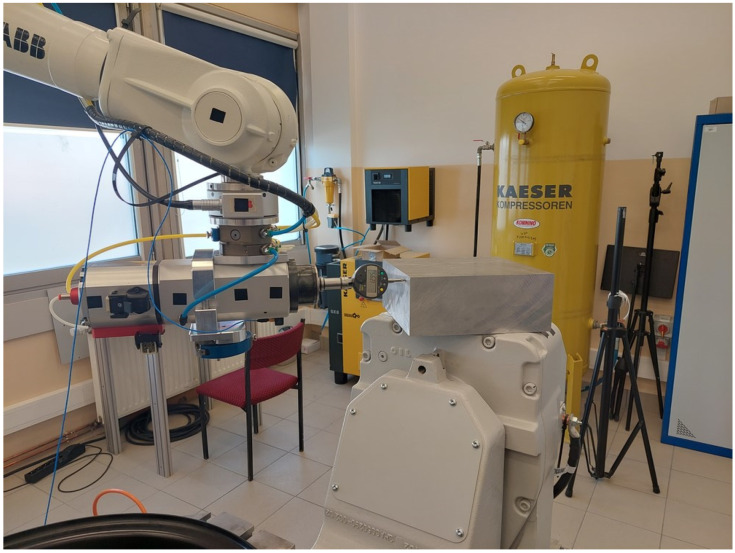
Robotic stand for determining the error in conditions similar to real ones.

**Figure 19 sensors-22-06464-f019:**
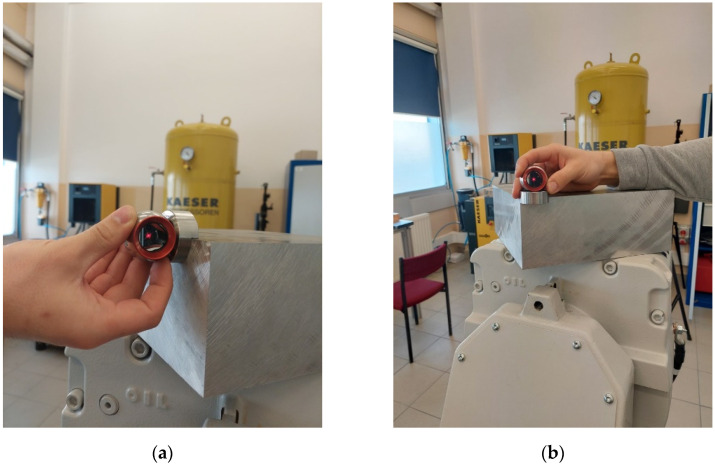
Determining the point using a retroreflector: (**a**) determining point 1; (**b**) determining point 2.

**Figure 20 sensors-22-06464-f020:**
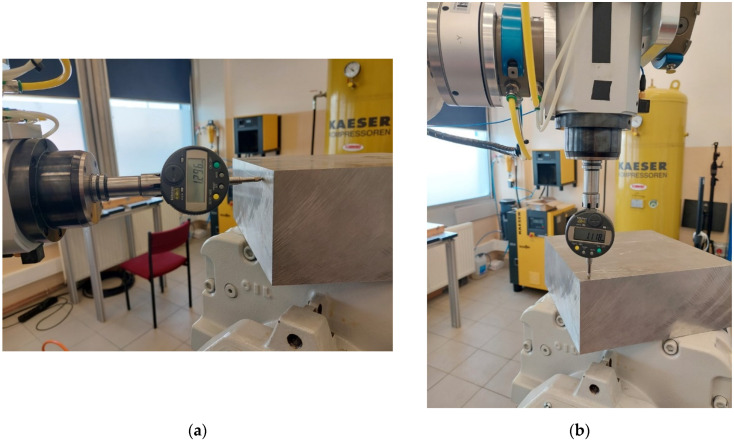
The movement of the robot’s TCP to the defined points: (**a**) TCP movement to point 1; (**b**) TCP movement to point 2.

**Table 1 sensors-22-06464-t001:** The maximum permissible error of the Leica Absolute Tracker AT960.

Laser Tracker Subsystem	Symbol	Maximum Permissible Error (MPE)
Interferometer (IFM)	^e^IFM	±0.4 μm + 0.3 μm/m
Absolute distance measurement (ADM)	^e^ADM	±10 μm
Parameter R0 (R0)	^e^R0	±3 μm
Transverse	^e^T	±15 μm + 6 μm/m

**Table 2 sensors-22-06464-t002:** Maximum permissible error.

Two-Face Measurement	Position	Tolerance (MPE)	General Formula
Absolute Angular Performance ^e^T	Pos. 1 to 9 (1.5 m distance)	±24 μm	±15 μm + 6 μm/m
	Pos. 10 to 18 (6 m distance)	±51 μm	

**Table 3 sensors-22-06464-t003:** “Birth certificate” of the owned IRB 2400 robot.

Accuracy Information from Verification	
Measures verification points	50
Average Absolute Error	0.18 mm
Maximum Absolute Error	0.38 mm
Standard Deviation	0.07 mm
Within Specification (<1 mm)	100 %

**Table 4 sensors-22-06464-t004:** Comparison of the robot programming methods.

Criterion	On-Line Programming	Off-Line Programming	Programming of Industrial Robots Using a Laser Tracker
Additional hardware and software requirements (in addition to the standard robot with a handheld controller)	None.	They are present. Required: CAD model of the robot, position and workpiece, dedicated software, additions increasing the absolute accuracy of the robot.	They are present. Required: laser tracker, accessories increasing the absolute accuracy of the robot.
Precision in pinpointing path points	High. It uses the repeatability of the robot. It is based on the programmer’s senses.	Medium. It uses the absolute accuracy of the robot.	Medium. It uses the absolute accuracy of the robot.
Programming time of path points	Long. Programming requires a lot of time and skill of the programmer. Travel times to points limited for manual operation 250 mm/s. In fact, access in close proximity to the point is carried out at speeds below 1 mm/s.	Short. Programming points is quick and consists in indicating them on the CAD model.	Short. Programming the points is quick and consists in indicating them with a retroreflector.
Level of programmer’s safety	Low. The programmer must be in the robotic station and directly control the robot. He or she is at risk of injury related to the robot or its accessories. He or she is exposed to noise. For safety reasons, it is recommended to program robots in groups of 2, which increases costs.	High. Most of the programmer’s work is performed on a PC, which may be remote from the robotic station.	High. After determining the robot’s base system, it can be turned off, increasing safety and reducing noise.
Safety of the station and the workpiece (possibility of damaging the robot, station and workpiece)	Low. During programming, the robot can damage the tool, station components or the workpiece.	High. The risk of damage only occurs during path tests.	High. The risk of damage only occurs during path tests.
Costs	Low. The purchase of additional equipment is not required. The greatest costs are generated by the programmers’ working time.	Medium. It is necessary to digitize the robot, position and workpiece. Costly off-line programming software must be purchased.	High. It is necessary to purchase or rent a laser tracker. It should be noted that universities or integrators often already have such equipment for other applications.

## Data Availability

The data are contained within the article.

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
