# Peer review of "Programming of Industrial Robots Using a Laser Tracker"

_sensors, 2022, doi:10.3390/s22176464_

Round 1

Reviewer 1 Report

This paper provides detailed background information, adequately describes an industrial robot programming method based on a laser tracker, and points out the advantages, disadvantages, and errors of this method, which contributes to improving the programming speed and safety of the operator. However, some problems still exist. 

(1) There are some nonstandard formats in this paper, such as the "TCP" used in many places does not introduce the full name.  

(2) The size, layout, meaning, expression, and connection between the picture need to be modified. For example, Figure 1 and Figure 2 are repeated, figure 1 can be deleted or the two can be combined. The description of Figure 3 and Figure 4 is not clear enough, so it is recommended to add notes. Part of the content in Figure 6 is irrelevant to this study, and it is suggested to organize it into a table. The description of Figure 7 is confusing and difficult to understand. It is suggested to advance the description of Figure 13 on the robot pedestal. figures in Figures 16 and 22 are of different sizes, and the content in the figure is not clearly described. Figure 19 is not clear, so it is suggested to replace it with a table. Pay attention to the connection between Figure 20, figure 21, and Figure 22, and strengthen the preciseness of the picture layout. The picture drawing on this paper is not beautiful enough.

(3) In the measurement of Figure 22, only two data obtained show errors of 0.296mm and 0.118mm. It is suggested to increase the measurement data to obtain more reliable measurement results.  

(4) The description of the robot station in this paper needs to be condensed to strengthen its relevance to this paper. 

(5) The advantages of this method in programming speed and security are mentioned in this paper, but there is a lack of quantitative comparison with other programming methods in the experiment. 

Reviewer 2 Report

This article introduces a new programming method based on the use of a laser tracker. This method is interesting, but has some problems. This method was not compared with other methods. At the same time, there are many problems with the format of the article. Therefore, I think it should be modified to meet the standard requirements of "sensors", and the specific modification suggestions are as follows:

1. The format of the article paragraph needs to be further modified, such as indenting the first line by 2 characters, etc.

2. The figure in the article should be centered.

3. The title numbers in the article should be further revised, and different titles should be marked with numbers.

4. The introduction in the introduction is too cumbersome, and the word "review" at the end should be replaced.

5. The title of the table in the text should be centered.

6. Missing comparisons in analysis. For the huge advantages mentioned in the conclusion, the necessary explanation is missing in the text. The authors please provide further explanation.

Reviewer 3 Report

General comments

=============

This paper explores using a laser tracker for specifying paths industrial robots will follow programatically.  In the developed method, the programmer indicates the path points with the help of a retroreflector and then robot follows the path. The programming of the paths is in part automated. The approach is compared against existing alternatives.

Specific comments

=============

Major comments

---------------------

(1)  The approach is qualitatively compared against existing alternatives but it is not actually evaluated against any other state of the art approach. An actual evaluation demonstrating the performance of the proposed approach against other existing approaches accomplishing the same task needs to be done to demonstrate effectiveness and success. Without this element it is very difficult to gauge if the research done is a contribution worthy of publication and one that the Sensors readership will benefit from.

(2) The study would be improved by identifying exactly who the audience is for this work. It is currently unclear who in the Sensors readership would benefit from this contribution and what are the actionable outcomes for the readers. The source code and data are not supplied and the reference list and context for the study is lean.

Identifying a specific audience or workflows within the community that this approach could be plugged into so that it would be actionable would demonstrate to readers how this work translates into practice and make the paper more impactful (and likely increase its future citation count).

In addition, there seems to be a lot of related work that would help set the context of this study missing. At the least I would have expected the following 3 pieces of research to be referenced.

Yu, Yongtao, et al. "Automated extraction of urban road facilities using mobile laser scanning data." IEEE Transactions on intelligent transportation systems 16.4 (2015): 2167-2181.

Li, Jing, et al. "Real-time self-driving car navigation and obstacle avoidance using mobile 3D laser scanner and GNSS." Multimedia Tools and Applications 76.21 (2017): 23017-23039.

Ferreira, Marcos, et al. "Stereo-based real-time 6-DoF work tool tracking for robot programing by demonstration." The International Journal of Advanced Manufacturing Technology 85.1 (2016): 57-69.

(3) In the replication crisis error an anonymized version of the data provided in the paper, the scripts used for analysis, and the scripts used to create the figures for the paper need to be provided to both the reviewers and to the readership. This ensures completely transparent analysis and makes the authors paper significantly more impactful as other researchers can build off it.

=============

Minor comments

---------------------

(4) In figures 2, 5, 8, 9, 10, 13 in the paper use red and green in contrast to one another. This is an unsafe color blind palette for 5% of the population that experiences red-green color blindness. Using a color-blindness safe color palette (https://davidmathlogic.com/colorblind/#%23D81B60-%231E88E5-%23FFC107-%23004D40) and larger text fonts would improve the readability of figures. 

(5) The data in Table 1 is centered and difficult to read it needs to be aligned left or right to make comparison of data items easier for readers.

(6) Within the introduction section of the paper, during the brief section where related research is discussed, the authors repeatedly use the phrase, "the paper" and then cite a resource. This can be improved to make for more pleasant reading by either describing the the actual work as opposed to referring to it as the paper. Or using the authors last name "Doe and Brown" instead of repeatedly using "The paper".

Round 2

Reviewer 1 Report

The paper is revised based on the comments. I accept this paper.

Please organize Fig5 in V2 into a table, cause the relevant data in /"Birth certificate" of the owned IRB2400 robot/ are helpful in this paper, do not let authors search data in the certificate, you should present them obviously.

Reviewer 3 Report

The paper is significantly improved. However, there are several minor issues that remain unaddressed.

(1) The authors provide the source code as a text file. This is improved but I still think in the replication replication crisis the source code (ending in a file extension appropriate to the programming language) should be provided. Also, the data presented in each table and any scripts used to generate the data should also be provided.  

(2) In addition I think there was a misunderstanding related to use of "the paper". My suggestion was to use the actual names of the authors of the referenced papers. It was not to use the phrase "the author" as opposed to "the paper". Using the authors names enables readers to see what work is referenced without having to scan until the final section of the paper.
